# Learning Transferable Policy Capabilities Through Cross-Domain Simulation RL

Yuqing Xie*, Yuxi Hong†, Feng Gao*, Mengyuan Fan‡, Kang Chen‡, Mingjie Wei†
Quanlu Zhang§, Ya Zhang¶, Chao Yu*, Yu Wang*

*Tsinghua University; †Harbin Institute of Technology; ‡Peking University; §Infinigence AI; ¶Shanghai Jiao Tong University

*Abstract*—**Large-scale vision-language-action (VLA) models promise efficient policy learning through reusable pretrained representations, but still require post-training for reliable closed-loop behavior. Existing reinforcement learning (RL)-based post-training methods often rely on high-fidelity simulators tailored to the deployment setting, which are expensive to build and still face sim-to-real gaps. We propose *cross-domain simulation RL*, a scalable paradigm for improving transferable policy capabilities using diverse source domains. A pretrained VLA policy is post-trained with supervised fine-tuning (SFT) warmup and RL in source simulators, then adapted to the target domain with limited demonstrations. Our method converts diverse simulators into transferable policy capabilities, improving both target-domain performance and adaptation efficiency. In sim-to-sim experiments, RL on two source domains improves the target-domain success rate by 25 points over the target-domain SFT policy, and single-source RL requires $32\times$ less target-domain data to reach the same target-domain performance. Real-robot results further show that source-domain RL can transfer to physical deployment, with the clearest gains appearing on tasks requiring precise manipulation.**

## I. INTRODUCTION

Robot foundation models aim to endow embodied agents with reusable perception, language understanding, and control capabilities. Vision-language-action (VLA) models [30, 13, 2, 23], and more recently world-action modeling approaches [44, 12], learn policies from broad robot data and multimodal observations. Ideally, such models should solve new tasks with little additional training. In practice, however, they remain far from zero-shot deployment. Their performance degrades under changes in task, scene, object distribution, embodiment, or sensing conditions [50, 16, 38, 27, 48, 6]. Post-training therefore remains critical.

Existing post-training methods mainly use supervised fine-tuning (SFT) on demonstrations or reinforcement learning (RL) through interaction. SFT is stable but bounded by demonstration coverage. RL improves closed-loop behavior through trial-and-error feedback [18, 22, 47], sometimes with world models [49] or preference learning [9], but requires interaction data that is expensive and risky on real robots. Simulation offers scalable interaction, automatic reward evaluation, and safe exploration.

Despite its promise, simulation RL usually assumes access to a simulator tailored to the target setting. Building such simulators for every robot, task family, object distribution, or deployment environment is costly, and target environments may be only partially specified before deployment. Meanwhile, many high-quality simulators already exist. This raises a central question: *can existing simulators improve policy learning in a novel target domain, even without RL in that target domain?*

We propose *cross-domain simulation RL*, a post-training paradigm for learning transferable capabilities from diverse source simulation domains. A policy first undergoes RL with SFT warmup across source domains, then adapts to an unseen target domain with limited demonstration data. Rather than treating simulation as a replica of the target environment, we use it as a scalable source of interaction signals for learning reusable closed-loop control structure.

We evaluate this pipeline in sim-to-sim and sim-to-real settings using OpenVLA-OFT and $\pi_{0.5}$. In sim-to-sim transfer, one-source RL improves target success by 15 points over target-only adaptation, while two-source RL increases the gain to 25 points, showing that broader source coverage yields stronger priors. Beyond improving final performance, the source-domain RL policy also improves data efficiency, reaching the same target-domain performance with $32\times$ less target-domain data than the base training pipeline.

We further validate the pipeline on real-robot tasks. After post-training in simulation, we transfer policies to physical deployment with limited real demonstrations and find that the most visible gains appear on tasks requiring precise closed-loop execution. These results support our hypothesis that cross-domain RL can strengthen policy capabilities that transfer across domain boundaries.

Our contributions are threefold.
- We introduce *cross-domain simulation RL* as an RL-based post-training paradigm for VLA models, improving policy learning in novel target domains without target-domain RL or target-specific simulation.
- We show in sim-to-sim experiments that source-domain RL transfers capabilities to unseen targets, improving target-domain performance while reducing the need for target-domain data.
- We show on real robots that cross-domain simulation RL can transfer to physical tasks, with the largest gains appearing on tasks requiring precise closed-loop execution.

Together, these results support reusing diverse simulations to build stronger embodied policies.

## II. RELATED WORK

### A. Reinforcement Learning for VLA Models

Recent work explores reinforcement learning as a post-training strategy for improving VLA performance and gen-

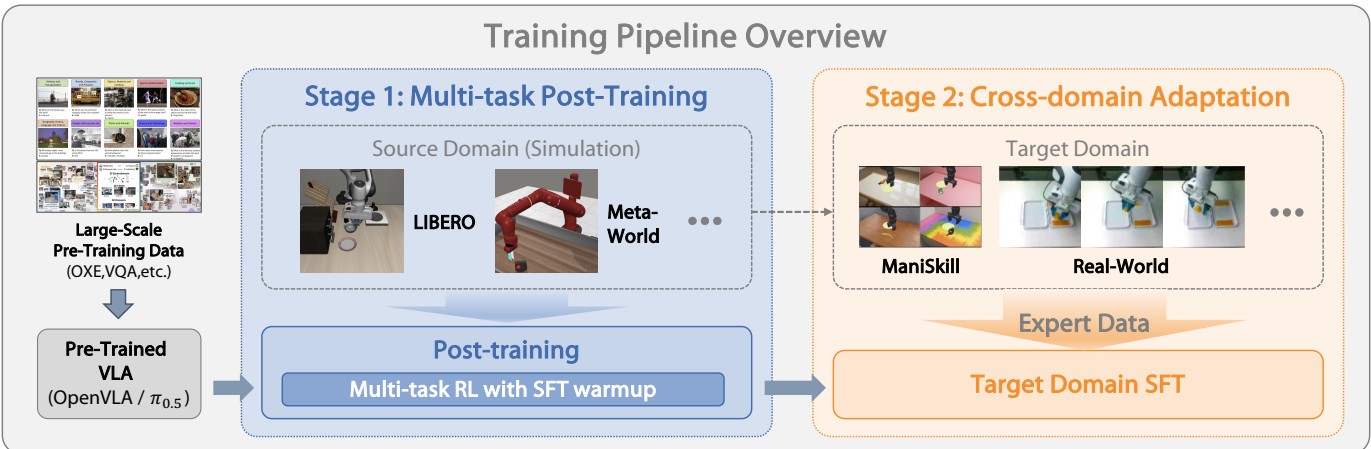

Fig. 1: Overview of our cross-domain simulation RL pipeline. **Stage 1 (Left):** Starting from a pre-trained VLA model, we perform multi-task RL on diverse simulation benchmarks using SFT as warmup. **Stage 2 (Right):** We adapt the policy to an unseen target domain using target-domain demonstration data.

eralization. Unlike supervised fine-tuning, RL optimizes task-level rewards through interaction and can better handle long-horizon errors and distribution shift. Online methods design dense or process-level rewards [5] and scalable RL systems for different VLA backbones [18, 3, 47, 22, 24]. Offline alternatives improve policies from fixed robot datasets by learning value or advantage estimates over demonstrations, autonomous rollouts, and interventions [10, 36], or by converting learned rewards into preference pairs for direct policy optimization [9, 41]. For flow-based VLAs, recent methods further balance advantage signals with flow-matching stability [25].

Most existing post-training studies still fine-tune and evaluate VLAs under the same task or domain distribution. MetaVLA [17] studies multi-task training across LIBERO tasks but remains restricted to a fixed task family, whereas we study whether source-domain RL improves few-shot adaptation to unseen targets.

*B. Sim-to-Real Transfer*

Robotic policies must operate in real environments, but physical interaction data is costly. Classical sim-to-real methods address the reality gap through domain randomization or high-fidelity digital twins that match real geometry, appearance, and dynamics [39, 28, 19, 40]. Recent work also explores sim-real co-training with simulated and real demonstrations [34]. In this work, we do not build a target simulator, tune simulation parameters, or run RL on the real robot. Instead, we apply RL in multiple source simulators to improve real deployment after few-shot adaptation.

*C. Multi-task Reinforcement Learning*

Multi-task RL trains one policy across multiple tasks [35], which aligns naturally with the training goals of VLA models. For smaller policies, direct multi-task RL often suffers from task interference and imbalance [14]: gradients may conflict, reward scales may differ, and easy tasks may dominate. Prior work addresses these issues with gradient surgery [46, 20, 1],

task scheduling [29, 4], task-specific modulation [32, 42], or soft parameter sharing [31, 43]. While these works focus on smaller policies trained from scratch, the multi-task recipe for pretrained VLAs with strong knowledge priors remains unclear. In this work, we find that a simple task sampler with standard RL is sufficient for cross-domain RL.

## III. CROSS-DOMAIN SIMULATION RL

We propose a two-stage cross-domain simulation RL framework, as shown in Fig. 1. Stage 1 learns capabilities exposed by source simulators, such as closed-loop correction, grasp timing, and action scaling. Stage 2 aligns these capabilities with target observations, dynamics, and action scales. We first warm up the policy with SFT on demonstrations, then perform multi-task RL in source simulations. The resulting checkpoints are adapted to a new target domain via few-shot SFT. The goal is not to reproduce the target domain, but to reuse source interaction to improve transferable closed-loop behavior. We describe the implementation using OpenVLA-OFT [15] as an example; $\pi_{0.5}$ [11] details are provided in Appendix A.

*A. Stage 1: Multi-task Post-training*

*1) Multi-task Supervised Fine-Tuning:* Our multi-task SFT follows standard VLA fine-tuning practices using behavior cloning. Each sample is a tuple $(o, l, a)$ of visual observation, language instruction, and expert action sequence.

OpenVLA-OFT adapts OpenVLA-7B to continuous action chunks $a = [a^{(1)}, ..., a^{(H)}] \in \mathbb{R}^{H \times D}$ with parallel decoding. We use the standard L1 regression objective over the predicted chunk:

$$\mathcal{L}_{\text{SFT}}^{\text{OFT}} = \frac{1}{H} \sum_{h=1}^{H} \|a^{(h)} - \hat{a}^{(h)}\|_1, \tag{1}$$

where $\hat{a}^{(h)}$ is the model prediction.

**Multi-task data mixing.** Each batch samples trajectories uniformly from source datasets:

$$\mathcal{B} = \bigcup_{i=1}^{K} \mathcal{B}_i, \quad |\mathcal{B}_i| = \frac{B}{K}, \tag{2}$$

where $B$ is the total batch size and $\mathcal{B}_i$ is the sub-batch from environment $i$. This uniform sampling strategy prevents large datasets from dominating gradient updates and ensures balanced learning across tasks.

**Environment-specific normalization.** Because source environments differ in action and state spaces, we maintain independent normalization statistics for each environment $\mathcal{E}_i$ during data loading and model decoding. This environment-specific normalization allows the model to operate within a consistent numerical range while preserving the relative structure of each environment's action distribution.

*2) Multi-task Reinforcement Learning:* For RL fine-tuning, we build on RLinf [45, 47] and implement a multi-environment parallel pipeline with GRPO and a simple task sampler. Source simulators differ in reward scales and control distributions, so the RL stage must extract reusable behavior without letting one domain dominate optimization. The pipeline runs multiple source tasks in parallel and updates one shared VLA policy, exposing it to diverse visual layouts, rewards, and manipulation behaviors.

**Problem formulation.** Each manipulation task is a POMDP $(\mathcal{S}, \mathcal{A}, \mathcal{T}, \mathcal{R}, \mathcal{O}, \Omega, \gamma)$. The agent observes $o_t \in \mathcal{O}$ and instruction $l$, outputs action chunk $a \in \mathbb{R}^{H \times D}$ via $\pi_\theta(a|o_t, l)$, and maximizes

$$J(\theta) = \mathbb{E}_{\tau \sim \pi_\theta}\left[\sum_{t=0}^{T} \gamma^t r_t\right]. \tag{3}$$

**GRPO** [7]. For each observation $o_t$, we sample $G$ action chunks, execute them, and collect returns $\{R^{(1)}, ..., R^{(G)}\}$. Advantages are normalized within the group:

$$A^{(g)} = \frac{R^{(g)} - \text{mean}(\{R^{(1)}, ..., R^{(G)}\})}{\text{std}(\{R^{(1)}, ..., R^{(G)}\}) + \epsilon_c}, \tag{4}$$

where $\epsilon_c$ is for numerical stability. This makes advantages less sensitive to reward-scale differences across environments, which is useful when source benchmarks define rewards differently. Therefore, we adopt GRPO for cross-domain RL.

The policy is optimized via the clipped surrogate objective:

$$\mathcal{L}_{\text{GRPO}}(\theta) = -\mathbb{E}\left[\min\left(\rho_t(\theta)A, \text{clip}(\rho_t(\theta), 1-\epsilon, 1+\epsilon)A\right)\right], \tag{5}$$

where $\rho_t(\theta) = \frac{\pi_\theta(a|o_t, l)}{\pi_{\theta_{\text{old}}}(a|o_t, l)}$ is the importance ratio and $\epsilon$ is the clipping parameter. The clipping constraint prevents overly large policy updates that can destabilize VLA fine-tuning.

**Task sampling.** At each iteration, we sample parallel rollouts across $K$ environments and bias sampling toward unsolved tasks with $p_i \propto (1 - S_i)^\alpha$, where $S_i$ is task success rate of task $i$ and $\alpha$ is the sampling temperature. Lower-success tasks are sampled more often, forming an automatic curriculum. We ablate this choice in Sec. IV-E2.

### B. Stage 2: Cross-domain Adaptation

After Stage 1, we adapt the policy to an unseen target domain using a small target-domain demonstration set. This stage aims to preserve transferable policy priors learned from source-domain RL while aligning the policy with target-domain visual and dynamics characteristics. We use the same SFT objective as in Stage 1 and keep environment-specific normalization for the target domain to ensure consistent action scaling. This adaptation stage is critical in our pipeline, as validated by the zero-shot ablation in Sec. IV-E1.

## IV. SIM-TO-SIM CROSS-DOMAIN TRANSFER

Following the training pipeline in Sec. III, we evaluate whether source-domain post-training improves adaptation to an unseen simulation target. We first compare post-training paradigms under fixed target-domain data budgets (Sec. IV-B), then measure target-data efficiency (Sec. IV-C) and test whether the trend holds across VLA architectures (Sec. IV-D). Sec. IV-E ablates Stage 2 SFT adaptation and Stage 1 multi-task RL algorithms. Appendix C provides additional trajectory visualizations, analysis of RL versus SFT contributions, robustness to target-domain perturbations, and results on the timing of few-shot adaptation.

### A. Experimental Setup

We use ManiSkill [37], LIBERO [21], and Meta-World [26] as benchmark tasks, with details in Table III. For the VLA models, we use OpenVLA-OFT and $\pi_{0.5}$. All runs start from the official pretrained checkpoints followed by the two-stage pipeline described in Sec. III. Unless otherwise stated, we report OpenVLA-OFT results. We compare **Base** (target-domain SFT only, without source post-training), **L-SFT/L-RL** (LIBERO SFT and SFT+RL), and **LM-SFT/LM-RL** (LIBERO+Meta-World SFT and SFT+RL). Training details are in Appendix B. We report task success following the official definitions and average over 256 episodes per task with fixed seeds.

### B. Simulation RL Improves Cross-domain Transfer

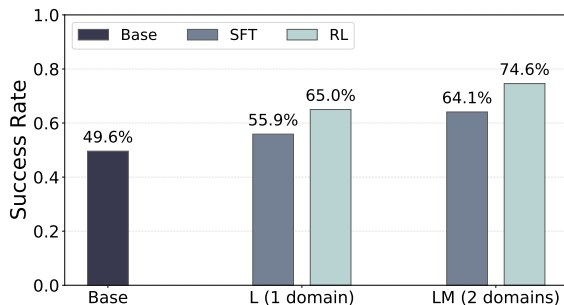

Fig. 2: Few-shot transfer performance scales with cross-domain simulation RL.

After source-domain post-training, we adapt each checkpoint to the target domain using target demonstrations. Here, ManiSkill serves as the unseen target domain, while LIBERO and Meta-World are used as source domains for post-training.

This setting mimics real-world deployment, where target demonstrations are costly, but target-domain RL is not. Fig. 2 shows three trends:

- **More source-domain diversity yields better transfer.** The Base model reaches 50% success after adaptation. Adding one source domain (L-RL) raises success to 65%, and adding two source domains (LM-RL) raises it to 75% (+25 points).
- **RL improves transferable capabilities beyond SFT.** RL outperforms its SFT counterpart in both settings (65.0% vs 55.9% for L, 74.6% vs 64.1% for LM), indicating that online interaction improves cross-domain transfer.
- **Scaling trends remain positive.** While SFT gains diminish as source domains increase, RL continues to provide improvements, suggesting transfer has not saturated with one source domain.

Overall, few-shot adaptation scales with both source-domain diversity and RL interaction, supporting the hypothesis that source simulators can improve capabilities useful outside their own domains.

### C. Sample Efficiency of Cross-domain Transfer

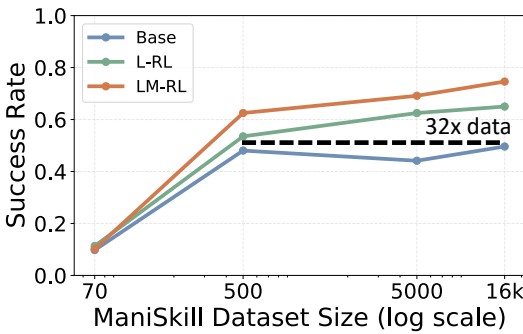

Fig. 3: Few-shot sample efficiency improves with cross-domain simulation RL.

To quantify target-domain sample efficiency, we vary the amount of target-domain demonstrations for adaptation. We fine-tune the models after simulation RL with 70, 500, 5K, and 16K trajectories from the ManiSkill dataset. Fig. 3 shows the result.

Transferable capabilities emerge in the 500-5K regime. At 500 trajectories, LM-RL reaches 62.5% and L-RL 53.5%, both well above Base (48.0%); at 5K, LM-RL and L-RL continue improving (69.1% and 59.3%) while Base drops to 44.1%, widening the gap to 25.0 points under the same adaptation protocol. At 16K, Base remains near 49.6%, whereas L-RL (63.8%) and LM-RL (74.6%) continue to climb, suggesting that source-domain training improves sample efficiency and raises the attainable performance ceiling in the target domain.

Comparing single-task (L-RL) and multi-task (LM-RL) post-training reveals consistent benefits from domain diversity. At 500 trajectories, LM-RL outperforms L-RL by 9.0 percentage points, and this gap persists even at 16K trajectories (+9.6 points), demonstrating that cross-domain learning provides benefits orthogonal to simply increasing data quantity within a single domain.

Additionally, we note that L-RL with 500 trajectories (53.5%) matches the Base model's performance with 16K trajectories (49.6%), translating into a $32\times$ reduction in target-domain data. This confirms that cross-domain simulation RL improves both adaptation quality and efficiency.

### D. Consistency Across VLA Architectures

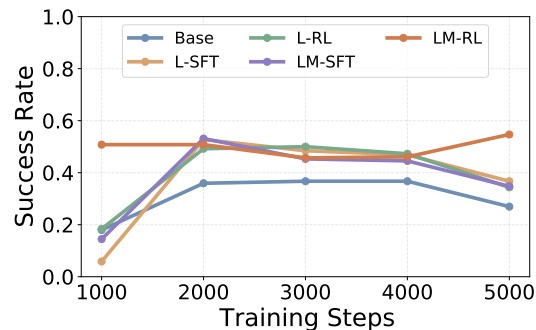

Fig. 4: Transfer scaling with $\pi_{0.5}$.

We repeat the few-shot transfer experiment with $\pi_{0.5}$, using ManiSkill as the target and non-ManiSkill environments as sources. Fig. 4 reports target-domain success over fine-tuning steps. LM-RL is the most stable checkpoint and reaches 55% success, while other variants peak earlier and then degrade. However, the absolute gain is smaller than for OpenVLA-OFT. A likely reason is that $\pi_{0.5}$ already produces better continuous actions through its flow-matching action head, making few-shot target adaptation more effective even without source RL. As a result, cross-domain RL still helps, but has less room to improve on ManiSkill tasks that mainly stress precise manipulation.

### E. Ablation Studies

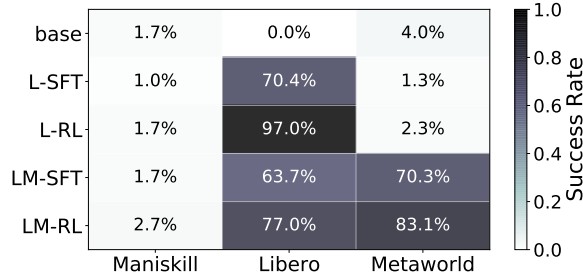

Fig. 5: Zero-shot cross-domain transfer.

*1) Ablating Stage 2: Cross-domain Adaptation:* We first ablate Stage 2 by removing target-domain adaptation and directly deploying post-trained policies to unseen target domains. In Fig. 5, rows denote post-training methods, and columns denote target domains used for evaluation. The results show that all models retain high success on seen domains but collapse on held-out domains. Stronger Stage 1, especially multi-domain RL, gives small zero-shot gains, but task performance remains far below few-shot adaptation and is not suitable for practical deployment. This ablation clarifies the role of each stage in our pipeline: Stage 1 improves capabilities, while Stage 2 performs

domain alignment. Their combination, rather than either stage alone, yields the strong few-shot transfer results in Sec. IV-B.

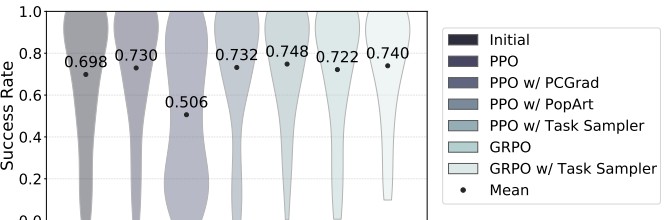

Fig. 6: Distribution of per-task success rates at step 300 across 50 tasks. For each method, the violin represents the task-performance distribution and the dot marks the mean.

*2) Ablating Multi-task RL Algorithm in Stage 1:* We next ablate multi-task RL choices on Meta-World MT50. We compare task sampling [4], PCGrad [46], PopArt [8], and two RL backbones, PPO [33] and GRPO. In this section, the base setup uses $\pi_{0.5}$ and PPO.

Fig. 6 reports the per-task success distribution at training step 300. PPO with task sampler achieves the highest mean success rate (74.8%), above vanilla PPO (73%) and the initial checkpoint (69.8%). PCGrad drops to 50.6%, with its distribution skewed toward low-success tasks, while PopArt (73.2%) provides only marginal improvement. These results suggest that a simple task selection mechanism is already effective for multi-task RL in VLA models.

PCGrad likely fails because each epoch is divided into multiple mini-batches with substantial gradient variance. The conflict-resolution projections in PCGrad amplify this noise, leading to conservative and unstable updates. PopArt normalizes value targets using task-specific statistics, but Meta-World v2 rewards are already tightly controlled across tasks, leaving little room for PopArt to help.

We also compare PPO and GRPO as RL backbones. GRPO with task sampler reaches 74%, slightly below PPO with task sampler (74.8%), likely due to lower sample diversity from grouped sampling. However, GRPO offers two advantages for large-scale training: (1) group-wise advantage normalization handles reward-scale mismatch across environments, and (2) removing the critic reduces compute overhead. Considering this trade-off, we adopt GRPO with task sampler as the default.

These results indicate that combining a simple task sampler with standard RL is sufficient for multi-task VLA training. Techniques designed for smaller models, such as PopArt and PCGrad, provide limited benefit when the model already encodes substantial prior knowledge.

## V. SIM-TO-REAL CROSS-DOMAIN TRANSFER

We evaluate cross-domain simulation RL on a physical Franka robot. We organize the real tasks along two axes: *manipulation precision*, where success requires a coherent sequence of accurate low-level actions, and *visual-language grounding*, where the policy must identify the correct object or color from the instruction before acting. To make manipulation precision diagnostic, *Pick yellow toy* uses a smaller object and

TABLE I: Real-robot evaluation settings. "Manip." denotes manipulation precision and "Ground." denotes visual-language grounding.

| Task | Manip. | Ground. | Description |
|---|---|---|---|
| Pick red toy | ✓ | | Pick and place the red toy into the plate. |
| Pick yellow toy | ✓ | | Pick smaller yellow toy from randomized starts. |
| Push red button | | ✓ | Push the red button among red and green buttons. |
| Push yellow button | | ✓ | Push the yellow button among red and yellow buttons. |

TABLE II: Real-robot success counts over 20 evaluation trials.

| Task | Base | L-SFT | L-RL | LM-SFT | LM-RL |
|---|---|---|---|---|---|
| Pick red toy | 0 | 11 | 20 | 18 | 19 |
| Pick yellow toy | 0 | 8 | 7 | 11 | 8 |
| Push red button | 16 | 13 | 17 | 17 | 14 |
| Push yellow button | 6 | 5 | 2 | 5 | 9 |

more randomized initial object positions. To test grounding beyond the seen instruction, we include *Push yellow button*, where policies are adapted with demonstrations for pushing the red button but are evaluated on pushing the yellow button.

**Protocol.** We use relative end-effector displacement control with wrist and front camera observations. For each task, we collect 50 few-shot demonstrations using a 3D space mouse for real-world adaptation, then evaluate each adapted policy over 20 trials.

**Manipulation-dominated tasks.** The clearest gain appears on pick-and-place tasks. The base OpenVLA-OFT policy fails on *Pick red toy* in all 20 trials, often oscillating the gripper near the object without forming a stable grasp. In contrast, L-RL reaches 20/20 success and LM-RL reaches 19/20, showing that cross-domain RL can substantially improve low-level action control. Qualitatively, RL-trained policies exhibit better end-effector alignment, grasp timing, placement accuracy, and recovery after imperfect approaches. The strongest transfer already appears after LIBERO RL, likely because LIBERO's tabletop view and camera geometry are closer to our real setup than Meta-World. The harder *Pick yellow toy* task is less conclusive. The smaller object and randomized starts increase localization and grasping difficulty, and the RL checkpoints do not consistently outperform SFT.

**Grounding tasks.** The grounding tasks show a different pattern. On the in-distribution *Push red button* task, the base policy already succeeds in 16/20 trials, and neither SFT nor RL provides a stable improvement. The out-of-distribution *Push yellow button* task is more revealing. LM-RL obtains the best result (9/20), while L-RL drops to 2/20. A plausible explanation is that adding Meta-World exposes the policy to a broader set of task-conditioned behaviors, whereas LIBERO-only RL provides little pressure to preserve color-conditioned selection. As a result, L-RL may overfit toward the adapted push-red behavior and degrade the capability needed for the held-out yellow-button instruction.

**Qualitative rollouts.** Fig. 7 provides representative rollouts

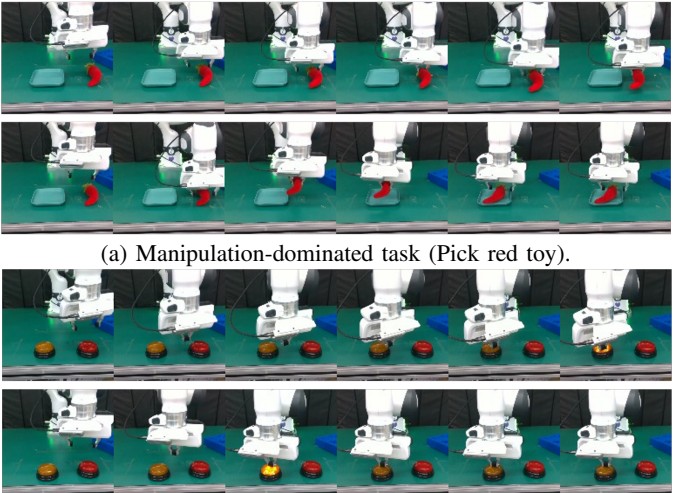

(a) Manipulation-dominated task (Pick red toy).

(b) Grounding task (Push yellow button).

Fig. 7: Real-world rollout visualizations. Each panel compares Base (top) and LM-RL (bottom).

from Base and LM-RL. On *Pick red toy*, the base policy often opens and closes the gripper beside the object, while LM-RL executes a cleaner approach and grasp. On *Push yellow button*, the failure modes are more nuanced. Base often identifies the correct region but fails to physically press the button, and even when it succeeds, it takes longer to finish. L-RL can execute the pressing motion but sometimes presses the wrong button. LM-RL is more likely to both reach the button and select the correct color. These rollouts are consistent with the quantitative trend. Source RL most clearly improves capabilities directly exercised by source domains, such as closed-loop manipulation precision, but does not automatically improve capabilities that are weakly represented in the source domains, such as color grounding.

Overall, the real-robot results show that cross-domain simulation RL can transfer useful policy capabilities to physical deployment. In our current tasks, the largest gains appear on tasks requiring precise closed-loop execution, while grounding tasks indicate that broader capability transfer depends on which capabilities the source domains exercise during post-training.

## VI. FUTURE WORK

This work is an in-progress study of cross-domain simulation RL for VLA post-training. Due to computational constraints, this work evaluates cross-domain simulation RL using only two source simulators. Scaling to a broader collection of simulators may further improve transfer, especially when source domains cover more diverse observations and task structures. Our results also suggest that transfer performance depends on the capabilities exercised by the source tasks. More diverse source domains may therefore be needed to transfer capabilities beyond closed-loop control, such as semantic understanding. Finally, our real-world evaluation is limited to a small number of tabletop tasks, and the current trials do not fully disentangle visual-language grounding failures from execution failures. A

more detailed diagnosis of which capabilities transfer, and how to design source tasks for specific target capabilities, remains an important direction for future work.

## VII. CONCLUSION

We presented *cross-domain simulation RL*, a post-training strategy for learning transferable policy capabilities from existing source simulators. In sim-to-sim transfer, two-source RL improves success by 25 points over target-only adaptation, and single-source RL matches the target-only baseline using $32\times$ less target data. Real-robot experiments further show that source-domain RL can transfer to physical deployment after limited real demonstrations, with the clearest gains on tasks requiring precise closed-loop execution. Together, these results suggest that existing simulators can serve as reusable post-training resources for improving robot policies without constructing a target-specific simulator.

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

## A. Supervised Fine-tuning

$\pi_{0.5}$ employs flow matching to generate continuous action chunks. It denoises standard Gaussian noise $z_0 \sim \mathcal{N}(0, \mathbf{I}_{H \times D})$ to the target action chunk $a \in \mathbb{R}^{H \times D}$ through a time-dependent velocity field $v_\theta(z_t, t, o, l)$, where $t \in [0, 1]$ is the flow time parameter. During training, we construct a linear interpolation path $z_t = (1 - t)z_0 + ta$ for $t \in [0, 1]$, where the true velocity along this path is $u_t = a - z_0$. The model learns to predict this velocity field via the conditional flow matching objective:

$$\mathcal{L}_{\text{SFT}}^{\text{Pi0.5}} = \mathbb{E}_{t, z_0, a} \left[ \|v_\theta(z_t, t, o, l) - (a - z_0)\|_2^2 \right]. \tag{6}$$

At inference, actions are generated by integrating the learned velocity field from $z_0$ to $z_1$ using the Euler method.

## B. Action Probability Computation for Reinforcement Learning

For OpenVLA-OFT, which predicts continuous actions via regression, we model the policy as a Gaussian distribution $\pi_\theta(a|o, l) = \mathcal{N}(a|\mu_\theta(o, l), \sigma^2 \mathbf{I})$, where $\mu_\theta(o, l)$ is the predicted action chunk and $\sigma$ is a learnable or fixed standard deviation. The log-probability is:

$$\log \pi_\theta(a|o, l) = -\frac{1}{2\sigma^2} \|a - \mu_\theta(o, l)\|_2^2 - \frac{HD}{2} \log(2\pi\sigma^2). \tag{7}$$

For $\pi_{0.5}$, which uses flow matching, we follow the approach in $\pi_{RL}$ [3] and compute action log-probabilities through the instantaneous change-of-variables formula. Given the ODE $\frac{dz_t}{dt} = v_\theta(z_t, t, o, l)$, the log-probability is:

$$\log \pi_\theta(a|o, l) = \log p_0(z_0) - \int_0^1 \nabla_{z_t} \cdot v_\theta(z_t, t, o, l) dt, \tag{8}$$

where $p_0(z_0) = \mathcal{N}(z_0|0, \mathbf{I})$ is the prior distribution, and $\nabla_{z_t} \cdot v_\theta$ is the divergence of the velocity field.

## C. Proximal Policy Optimization

PPO [33] is a widely used on-policy reinforcement learning algorithm that stabilizes policy updates via clipped importance sampling. For each observation $o_t$, the policy $\pi_\theta$ generates actions $a_t$, which are executed in the environment to collect rewards and form trajectories. The discounted return is defined as $R_t = \sum_{k=0}^{T} \gamma^k r_{t+k}$.

PPO relies on a learned value function $V_\phi(o_t)$ to estimate the advantage. In practice, the advantage $A_t$ is computed using generalized advantage estimation (GAE):

$$A_t = \sum_{l=0}^{\infty} (\gamma\lambda)^l \delta_{t+l}, \quad \delta_t = r_t + \gamma V_\phi(o_{t+1}) - V_\phi(o_t), \tag{9}$$

The policy is optimized using the clipped surrogate objective:

$$\mathcal{L}_{\text{PPO}}(\theta) = -\mathbb{E} \left[ \min \left( \rho_t(\theta)A_t, \text{clip}(\rho_t(\theta), 1 - \epsilon, 1 + \epsilon)A_t \right) \right], \tag{10}$$

where $\rho_t(\theta) = \frac{\pi_\theta(a_t|o_t, l)}{\pi_{\theta_{\text{old}}}(a_t|o_t, l)}$ is the importance sampling ratio between the current and old policies, and $\epsilon$ is the clipping parameter.

In addition, PPO jointly optimizes a value function loss and an entropy regularization term to improve stability and exploration:

$$\mathcal{L}_{\text{value}} = \mathbb{E} \left[ (V_\phi(o_t) - R_t)^2 \right], \quad \mathcal{L}_{\text{entropy}} = \mathbb{E}[-\mathcal{H}(\pi_\theta(\cdot|o_t))]. \tag{11}$$

The final objective combines these terms:

$$\mathcal{L} = \mathcal{L}_{\text{PPO}} + c_v \mathcal{L}_{\text{value}} - c_e \mathcal{L}_{\text{entropy}}, \tag{12}$$

where $c_v$ and $c_e$ are weighting coefficients.

### D. Classic Multi-task RL Implementations

We aim to identify an effective multi-task RL training procedure for VLA models without modifying the model architecture or introducing task-specific components. Therefore, in Sec. IV-E2, we focus on the following classic methods.

- **PCGrad** [46] is a gradient surgery method that addresses conflicting gradients and task interference. Let $g_i$ and $g_j$ denote the gradients of the shared parameters $\theta$ with respect to tasks $i$ and $j$. If the cosine similarity is negative, PCGrad modifies $g_i$ as $g_i \leftarrow g_i - \frac{g_i^\top g_j}{\|g_j\|^2} g_j$, if $g_i^\top g_j < 0$. This ensures that updates do not move shared parameters in directions that harm other tasks. We apply PCGrad to both actor and critic training.

- **PopArt** [8] normalizes the value function to handle heterogeneous reward scales. Each task has a value function $V_\phi(s)$ that predicts the task return $G_t$. PopArt maintains running estimates of the mean $\mu$ and standard deviation $\sigma$ of returns per task, normalizes targets, $\hat{G}_t = \frac{G_t - \mu}{\sigma}$, and then updates $V_\phi(s)$ using $\hat{G}_t$. After updating $\mu$ and $\sigma$, the output layer is rescaled to preserve the unnormalized value predictions.

- **Task Scheduler** selects tasks based on their learning progress. We implement a simple and straightforward variation. The probability of sampling task $i$ is $p_i \propto (1 - S_i)^\alpha$, where $S_i$ is the success rate of task $i$ and $\alpha$ is the sampling temperature. Tasks with lower success rates are sampled more frequently, forming an automatic curriculum.

## APPENDIX B
### DATASETS AND TRAINING PARAMETERS

TABLE III: Overview of benchmark tasks used in our experiments. Dataset size indicates the number of expert demonstration trajectories available for SFT training.

| Benchmark | # Tasks | Task Horizon | Dataset Size |
|---|---|---|---|
| ManiSkill | 16 | $\sim 31$ steps | 16000 demos |
| LIBERO-130 | 130 | $\sim 150$ steps | 5660 demos |
| Meta-World | 50 | $\sim 81$ steps | 2500 demos |

In this section, we provide comprehensive implementation details and hyperparameter configurations for both the supervised fine-tuning (SFT) and reinforcement learning (RL) phases across the evaluated vision-language-action (VLA) architectures.

### A. Hardware and Computation

All models were trained and evaluated on a high-performance computing cluster equipped with 8 NVIDIA A100 (80GB) GPUs. To optimize GPU memory usage and scale the training of large models, we utilized Fully Sharded Data Parallel (FSDP) with `bfloat16` precision during the RL fine-tuning of OpenVLA-OFT. For the $\pi_{0.5}$ model, standard data parallelism without sharding was sufficient under our specific batch size configurations. The multi-task SFT phase took approximately 12 hours, while the RL fine-tuning phase took around 40 hours per run.

### B. Supervised Fine-Tuning (SFT) Configurations

Both OpenVLA-OFT and $\pi_{0.5}$ are fine-tuned for a consistent duration of 30,000 steps. Detailed hyperparameter configurations for simulation and real-world adaptation are summarized in Table IV and Table V. For OpenVLA-OFT, we additionally employ image augmentation to enhance representation robustness and prevent overfitting.

Table IV and Table V summarize the detailed SFT hyperparameters for OpenVLA-OFT and $\pi_{0.5}$, respectively.

TABLE IV: SFT hyperparameters for OpenVLA-OFT.

| Parameter | Value | Parameter | Value |
|---|---|---|---|
| Base Model | OpenVLA-7B | Batch Size | 12 |
| Action Head | L1 Regression | Training Steps | 30,000 |
| LoRA Rank | 32 | Image Augmentation | True |
| LoRA Dropout | 0.0 | Num Images | 2 |
| Learning Rate | 5e-4 | Real Dataset | shuangqing_real_dataset |

### C. Reinforcement Learning (RL) Configurations

For the multi-task RL post-training, we primarily utilize the group relative policy optimization (GRPO) algorithm. The hyperparameters, detailed in Table VI, were determined based on a combination of official repository recommendations and empirical searches.

Notably, we adapted several hyperparameters to suit the distinct architectures. For $\pi_{0.5}$, we used a lower Adam $\beta_2$ value (0.95) and chunk-level log-probability computation to better stabilize the flow-matching training process. During evaluation, we applied a lower sampling temperature ($T = 0.6$) for $\pi_{0.5}$ to reduce the variance of the generated action chunks. In contrast, for OpenVLA-OFT, token-level log-probability and a standard $\beta_2$ (0.999) yielded the best convergence.

TABLE V: SFT hyperparameters for $\pi_{0.5}$.

| Parameter | Value | Parameter | Value |
|---|---|---|---|
| Model | Pi0.5 (pi05=True) | Batch Size | 256 (8 GPUs) |
| PaliGemma | gemma_2b | Learning Rate | 5e-5 (Cosine Schedule) |
| Action Expert | gemma_300m | LR Warmup Steps | 10,000 |
| Action Horizon | 10 | LR Decay Steps | 1,000,000 |
| Action Dim | 32 | LR End | 5e-5 (Constant) |
| Discrete State Input | False | Optimizer | AdamW ($\beta_1 = 0.9, \beta_2 = 0.95$, |
| Max Token Len | 200 | | weight_decay=1e-10) |
| Image Size | 224×224 | Gradient Clip Norm | 1.0 |
| Precision | bfloat16 | EMA Decay | 0.999 |
| Weight Init | pi05_base | Training Steps | 30,000 |

TABLE VI: Reinforcement Learning Hyperparameters Comparison.

| Parameter | OpenVLA-OFT | $\pi_{0.5}$ (OpenPI) |
|---|---|---|
| Algorithm | GRPO | GRPO (or GAE) |
| Num Action Chunks | 8 | 5 |
| Num Steps | — | 5 |
| Actor Learning Rate | 2e-5 | 5e-6 |
| Value Learning Rate | 3e-3 | 1e-4 |
| Adam $\beta_2$ | 0.999 | 0.95 |
| Reward Type | Action-level | Chunk-level |
| Logprob Type | Token-level | Chunk-level |
| Clip Ratio | 0.28 | 0.2 |
| Temperature | 1.6 | 1.0 (train) / 0.6 (eval) |
| Top-$k$ | -1 (Disabled) | 50 |
| Global Batch Size | 16,384 | 32,768 (2 envs) / 2,048 (3 envs) |
| Gradient Checkpointing | True | False |
| Sharding Strategy | FSDP | No Shard |
| Group Size | 8 | 8 |
| Reward Coef | 5.0 | 5.0 |
| $\gamma$ / GAE $\lambda$ | 0.99 / 0.95 | 0.99 / 0.95 |

## APPENDIX C
## FURTHER EXPERIMENTS ON SIM-TO-SIM CROSS DOMAIN TRANSFER

### A. Qualitative Analysis of Simulated Rollouts

To further illustrate the behavioral differences resulting from our post-training pipeline, we provide qualitative visualizations of robot rollouts in the ManiSkill simulation environment. We select a representative *PickAndPlace* task, where the robot must grasp a red bottle and place it onto a designated target plate. For each post-training checkpoint, we record a full evaluation episode and uniformly sample 18 frames to visualize the end-effector trajectory.

Fig. 8 compares the execution trajectories across four checkpoints: Base (directly adapted to ManiSkill), L-SFT, L-RL, and LM-RL. A clear progression in trajectory quality is visible. With SFT-only post-training (Base, L-SFT), the arm exhibits improved directionality and descends toward the table surface, but the end-effector positioning remains imprecise, leading to failed or unstable grasps. Incorporating RL (L-RL) yields a more purposeful approach trajectory with better alignment to the object, though occasional inaccuracies persist. The multi-task RL checkpoint (LM-RL) produces the most stable and precise trajectories. The robot smoothly approaches the bottle, executes a reliable grasp, and transfers the object toward the target plate.

These qualitative observations are consistent with the quantitative results reported in Sec. IV-B. The progressive improvement from Base to LM-RL mirrors the success-rate gains in the main experiments, confirming that multi-task RL post-training not only improves aggregate metrics but also yields visibly smoother and more reliable manipulation behaviors.

### B. Disentangling SFT and RL Data

The previous sections show that multi-task post-training substantially improves few-shot transfer performance. However, the relative contributions of SFT and RL during post-training remain unclear. In practice, RL post-training is typically initialized from an SFT checkpoint to accelerate convergence, which means that RL models are exposed to more data during post-training than SFT-only models. To enable a fair comparison, we design a data-controlled experiment.

Specifically, we compare two checkpoints: (1) LIBERO-Spatial SFT+RL: a model first fine-tuned with SFT and then further optimized with RL on LIBERO-Spatial (10 tasks); (2) LIBERO-130 SFT: a model trained using SFT only on the full LIBERO-130 dataset (130 tasks). Both models observe a comparable number of LIBERO trajectories during post-training, but the data distributions differ. The former acquires data through online RL interactions, concentrated within a narrow task domain, whereas the latter leverages more diverse offline demonstrations, spanning a broader set of tasks.

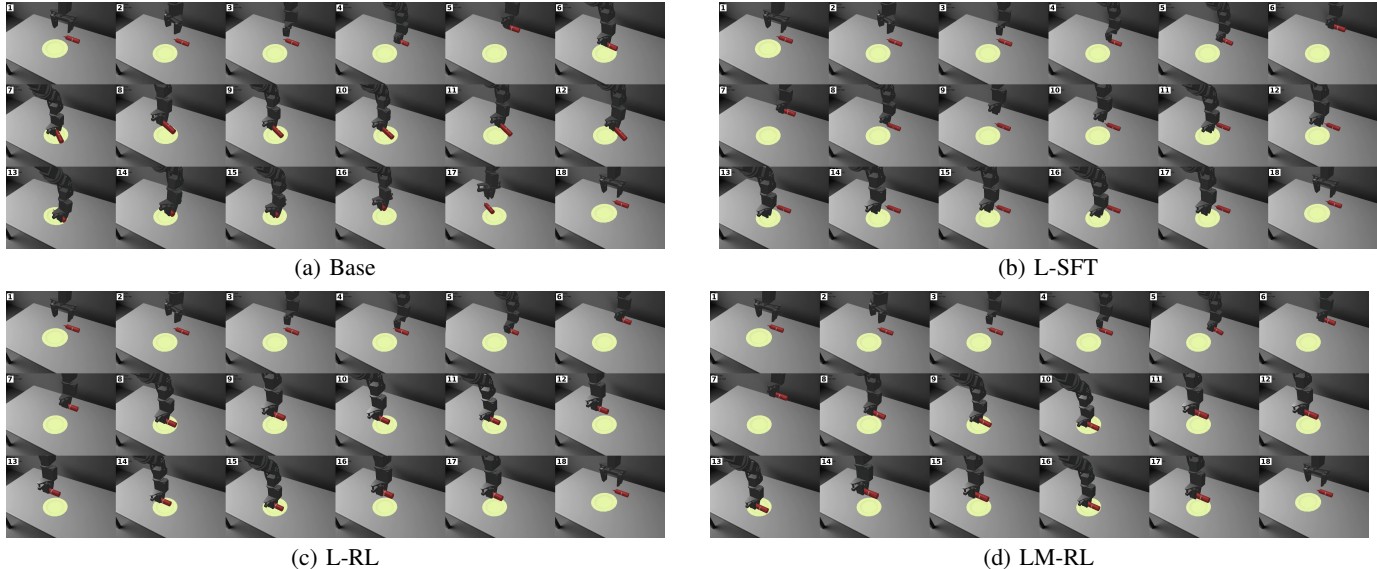

(a) Base        (b) L-SFT

(c) L-RL        (d) LM-RL

Fig. 8: **Visualization of simulated rollouts in ManiSkill across different post-training checkpoints.** Each panel shows 18 uniformly sampled frames (3 rows × 6 columns, read left-to-right, top-to-bottom) from a single episode of the *PickAndPlace* task. The robot must grasp a red bottle and place it onto the target plate. Trajectory quality progressively improves from Base to LM-RL, transitioning from erratic, off-workspace motions to smooth, task-completing behaviors.

After SFT adaptation on the ManiSkill domain, we evaluate both checkpoints on ManiSkill and find that LIBERO-130 SFT reaches a **55.86%** success rate, whereas LIBERO-Spatial SFT+RL reaches **65.0%**. Despite the greater task diversity in LIBERO-130 SFT, the LIBERO-Spatial SFT+RL model achieves a higher transfer success rate on ManiSkill. This corresponds to an absolute improvement of about 10%, suggesting that RL provides benefits beyond simply increasing data volume. By directly interacting with the simulator, RL enables the model to explore regions of the state space that are underrepresented in offline data, and thus mitigate overfitting to spurious visual patterns in demonstrations. Moreover, reward-driven optimization may implicitly promote the model to learn the shared structure across tasks.

## C. Transfer to Different Target Domains

To understand how multi-task post-training affects transfer to diverse target domains, we examine transfer performance from different post-training checkpoints to various target domains, including (1) from LIBERO RL post-training to ManiSkill, CALVIN, and Meta-World and (2) from LIBERO + Meta-World RL post-training to ManiSkill and CALVIN domains. We present the result in Table VII.

TABLE VII: Few-shot transfer success rates from different post-training checkpoints to target domains.

|       | ManiSkill | CALVIN | Meta-World |
|-------|-----------|--------|------------|
| Base  | 49.6      | 23.8   | 39.1       |
| L-RL  | 66.0      | 21.0   | 44.7       |
| LM-RL | 71.7      | 26.0   | –          |

For ManiSkill, both L-RL and LM-RL demonstrate strong positive transfer, with improvements of +16.4 and +22.1 percentages over the base, respectively. For Meta-World, L-RL provides modest gains (+5.6 points), indicating partial skill transferability from LIBERO.

Transfer to CALVIN reveals a more complex pattern. L-RL underperforms the base model, whereas LM-RL recovers from this negative transfer. The sequential tasks in CALVIN differ fundamentally from the manipulation tasks in LIBERO, and single-domain RL on LIBERO may reinforce task-specific behaviors that interfere with CALVIN's requirements. However, incorporating Meta-World alongside LIBERO provides a more balanced set of skills that generalizes better to structurally different domains like CALVIN.

These results suggest that effective post-training requires careful consideration of domain compatibility. Adding more source domains generally improves transfer, since diverse task structures can provide complementary benefits and reduce the risk of negative transfer to dissimilar targets.

## D. Robustness to Environmental Perturbations

Beyond cross-domain transfer, we investigate whether multi-task post-training improves robustness to environmental variations within a domain. We adopt the LIBERO-PRO benchmark [48], which systematically evaluates VLA models under controlled perturbations. We consider the following dimensions:

- **Semantic perturbations**, which vary language instructions via paraphrasing, synonym substitution, and syntactic restructuring.
- **Position perturbations**, which randomize the initial spatial layout of objects and goal locations, testing generalization beyond memorized configurations.
- **Object perturbations**, which modify visual attributes of manipulated objects (e.g., color, texture, size, and shape) while preserving task semantics.

We fine-tune a base VLA model on the LIBERO expert dataset under two settings: (1) **Base**, without additional post-training; (2) **M-RL**, multi-task RL on ManiSkill. We then evaluate all models on LIBERO-10 tasks under different LIBERO-PRO perturbation conditions.

TABLE VIII: Success rates on LIBERO-10 under different perturbation settings in LIBERO-PRO.

|  | LIBERO | Semantic | Position | Object | All |
|---|---|---|---|---|---|
| Base | 72.98 | 76.61 | 00.20 | 53.02 | 75.40 |
| M-RL | 82.06 | 85.89 | 00.00 | 64.31 | 86.49 |

We present the results in Table VIII. LIBERO denotes the original evaluation setting without perturbations, while **All** combines semantic, position, and object perturbations.

Multi-task RL post-training (M-RL) consistently improves performance on the original LIBERO setting and under semantic and object perturbations. In particular, M-RL achieves absolute improvements of +9.3% under semantic perturbations and +11.3% under object perturbations compared to the Base model, indicating enhanced robustness to linguistic variation and visual changes.

However, both models exhibit near-zero success under position perturbations, indicating a severe limitation in spatial generalization. Notably, RL post-training does not mitigate this failure mode, suggesting that current VLA models struggle to handle large changes in object layout and may overly rely on memorized spatial configurations.

## E. Timing of Few-shot Adaptation

Our previous experiments follow a two-stage paradigm: multi-task post-training on non-target domains, followed by few-shot adaptation to the target domain. Here, we test whether target-domain data can be incorporated earlier during multi-task post-training, and whether RL in non-target domains can directly improve target-domain performance without a separate adaptation step.

We compare two strategies using LIBERO as the primary post-training domain and ManiSkill as the target.

- **Late Adaptation**: We perform SFT and RL on LIBERO, then SFT the model with the ManiSkill dataset, and finally evaluate it on ManiSkill.
- **Early Adaptation**: We mix ManiSkill and LIBERO data during the initial SFT phase, perform RL only on LIBERO, and directly evaluate the model on ManiSkill without further adaptation.

Late Adaptation achieves a **65.0%** success rate, while Early Adaptation achieves only **2.3%**. This dramatic gap demonstrates that early inclusion of target-domain data is insufficient if followed by intensive training on other domains. The RL phase on LIBERO, which involves hundreds of gradient updates, overwrites the limited ManiSkill knowledge acquired during the initial mixed SFT, resulting in catastrophic forgetting.

We conclude that, for effective transfer, target-domain adaptation must occur after or throughout multi-task post-training. Simply mixing target-domain data in early training stages provides no benefit if the model subsequently undergoes extensive updates on other domains without maintaining target-domain exposure.