# OpenReview forum: "Learning Transferable Policy Capabilities Through Cross-Domain Simulation RL"
_roboticsfoundation.org/RSS/2026/Workshop/RL4VLA — RL4VLA_

### Official Review · Reviewer_P2Za · 2026-06-28
**Well-motivated and complete experiments, but the demonstrated claim is incremental.**

**Rating:** 5
**Confidence:** 5

**Review:**

The paper proposes cross-domain simulation RL, a scalable paradigm for improving transferable policy capabilities using diverse source domains in simulation. The two most common methods for VLA post-training: SFT and real-world RL are bottlenecked by demonstration coverage and costly rollouts respectively. The authors therefore ask whether simulators can improve a  policy's novel target domain performance without task-tailored simulators.

Relevance to workshop: Good fit showing the utility of using simulators for RL post-training as a mechanism for improving VLAs.

Strengths:
The central question of if off-the-shelf simulators can improve a novel target without building a target-specific simulator is well-motivated and practically appealing. Experiments are thorough and well executed.

Weaknesses:
The paper's central claim is that cross-domain RL pretraining yields transferable knowledge that improve downstream few-shot SFT. This is useful, but it is essentially the "better pretraining helps fine-tuning" claim. For example, a more interesting/novel claim could be zero-shot generalization to unseen tasks without target data, yet in Fig. 5, the paper shows that target data is necessary only reaching non-trivial performance with it.

The paper's named claim that source-domain RL transfers better than source-domain SFT (Sec. IV-B) is well-supported in the sim-to-sim experiments, but does not hold on the real robot experiments (e.g. L-RL beats L-SFT on Pick-red-toy 20 vs 11, but LM-SFT beats LM-RL on Push-red-button 17 vs 14). The real-robot prose sidesteps this by comparing RL to the base policy rather than to its SFT counterpart. Since the real robot is the deployment setting the paper is motivated by, the central "RL transfers better" claim holds only in the setting that ultimately matters least; an explanation for why the sim advantage vanishes on hardware should be included.

Phrasing:
"Few-shot" is a stretch given that the benefits emerge only at 500–5K demos (Fig. 3).
"RL backbones" should be "RL algorithms" as PPO and GRPO are algorithms.

---

### Official Review · Reviewer_ySU2 · 2026-07-01
**Review of Learning Transferable Policy Capabilities Through Cross-Domain Simulation RL**

**Rating:** 7
**Confidence:** 4

**Review:**

The authors propose cross-domain simulation RL, a pipeline for post-training vision-language-action models in simulation, without building a high-fidelity digital twin of the target setup.
The authors particularly propose using a set of simulated environments and tasks that are unrelated to the target domain for post-training the VLA both via supervised finetuning and multi-task RL. The experimental evaluation demonstrates that such post-training can improve policy transfer to the target setup in terms of efficiency of target demonstrations, as well as final performance.

Minor concerns:
	- the authors claim that building a high-fidelity digital twin is costly in terms of time and effort, but do not provide an explicit comparison in terms of performance when such target-domain simulator exists. Since maniskill is used and available as a target domain, this would seem like the natural oracle comparison to quantify the costs of avoiding building such high-fidelity simulator scene.
	- Table II shows that performance on pick-and-place tasks vastly increases with the proposed post-training method, but stays the same or roughly decreases for the "push" tasks. This raises the question on whether the post-training data is favorable only because it includes tasks that are similar as the target setup, and viceversa can harm when these are not similar.
	- How much does the choice of simulator tasks affect the final performance in the target domain? it seems to me that source vs. target similarity is an uncontrolled parameter that is not investigated quantitavely via a metric. In turn, this can cause the evaluation to be biased towards similarity and choices of post-training environments with respect to the target setup, rather than a general finding.

I praise the authors for the experiment displayed in Fig. 3, which nicely investigates how the various paradigms affect the performance of the policy in the target domain w.r.t. the amount of target-specific demonstrations. Interestingly, it seems that post-training via single-source or multi-source simulators increase the results at a higher data regime only. I believe this result shall be highlighted clearly, and probably deserves more investigation.

---

### Decision · Program_Chairs · 2026-07-03

**Decision:**

Accept

**Comment:**

This paper studies cross-domain simulation RL for post-training vision-language-action models without requiring a simulator for the target task. The reviewers found the problem important and liked the experimental results. The main concerns are about understanding why the method works, how the choice of source simulators affects transfer, and whether the claims about RL outperforming SFT are fully supported, especially on the real robot experiments. We believe that despite these concerns, the paper is a valuable contribution to the workshop. For the camera-ready version, the authors should clarify the discussion of the real-robot results, better explain when RL is expected to outperform SFT, and expand the discussion on the role of source-task similarity. In the future, a more systematic study of simulator similarity and transfer would be valuable.